# Ethnic Differences in Juvenile Idiopathic Arthritis in the Circumpolar Region

**DOI:** 10.3390/children12111525

**Published:** 2025-11-11

**Authors:** Sargylana G. Boeskorova, Marina V. Afonskaya, Vera M. Argunova, Polina A. Sleptsova, Liudmila V. Leonteva, Vasilina V. Nikiforova, Irina A. Chikova, Alexandr A. Yakovlev, Tatiana E. Burtseva, Mikhail M. Kostik

**Affiliations:** 1Department of Paediatrics and Children Surgery, Medical Institute, North-Eastern Federal University Named After M.K. Ammosov, Yakutsk 677000, Russiabourtsevat@yandex.ru (T.E.B.); 2Department of the Cardio Rheumatology of the Pediatric Centre of the Republican Hospital No. 1–M.E. Nikolaev National Center of Medicine, Yakutsk 677019, Russia; 3Department of Hospital Therapy, Occupational Diseases and Clinical Pharmacology, Medical Institute, North-Eastern Federal University Named After M.K. Ammosov, Yakutsk 677000, Russia; 4The Rheumatic Disease Center of the Clinic of the Yakut Science Centre of Complex Medical Problems, Yakutsk 677000, Russia; 5Hospital Pediatrics, Saint-Petersburg State Pediatric Medical University, Saint-Petersburg 194100, Russia; irinachikova@gmail.com (I.A.C.);; 6Laboratory of the Monitoring of Children Health and Medico-Environmental Research, Yakut Science Centre of Сomplex Medical Problems, Yakutsk 677000, Russia

**Keywords:** juvenile idiopathic arthritis, HLA B27 antigen, seronegative spondyloarthritis, entesitis-related arthritis, epidemiology, arthritis, Yakutia, Sakha, HLA-B27-associated arthritis, ethnic variation, arctic health, circumpolar health

## Abstract

**Highlights:**

**What are the main findings?**
Sakha children have a unique profile of JIA: higher prevalence, entesitis-related arthritis predominance, late access to biologic therapy, and lower probability to obtain remission with the first biological drug.

**What is the implication of the main finding?**
The optimization of the healthcare system: contemporary web-service technologies, including artificial intelligence, may shorten the gap to specialists’ consultation, and administration of the treatment, allowing entry into the window of opportunity and improving the disease’s outcomes (increasing the probability of remission).

**Abstract:**

Introduction: Rheumatic diseases, including spondyloarthritis, systemic lupus erythematosus, Takayasu’s nonspecific aortoarteritis, Behcet’s disease, and Kawasaki disease, are more prevalent among Asian populations. The indigenous Sakha people, who live in the harsh conditions of the North and the Arctic regions, exhibit a unique pattern of health issues. AIM: The objective of the study is to characterize the ethnic characteristics of juvenile idiopathic arthritis (JIA) among children from the indigenous population of the Republic of Sakha (Yakutia) and compare them with Caucasians (Russians) living in the same region. This comparison aims to inform the development of tailored diagnostic and treatment strategies. Methods: The comprehensive, single-center, retrospective cohort study included medical data of all Sakha (*n* = 168) and Russian ethnic patients (*n* = 48) with JIA who were examined and treated at the Pediatric Center’s Cardiorheumatology Department at the Republican Hospital No. 1–M.E. Nikolaev National Center of Medicine—between 2016 and 2023. The ethnicity was self-reported. The standard clinical procedures and laboratory assessments, as well as the current treatment regimen, were thoroughly reviewed. Results: It was found that children of Sakha descent had a later onset of juvenile idiopathic arthritis (JIA), which was associated with the enthesitis-related arthritis (ERA) categories (51.2% vs. 18.3%, *p* = 0.0002). They also exhibited higher prevalence of enthesitis (19% vs. 2.0%, *p* = 0.003), sacroiliitis (23.8% vs. 2.0%, *p* = 0.0003), and HLA-B27 antigen positivity (46.3% vs. 14.6%, *p* = 0.00005). The Sakha population exhibited a notably higher prevalence (41.7%) of ERA, compared to the Russian population (33.3%; *p* = 0.0003), and they initiated biologic therapy at a later stage. However, remission rates were lower among Sakha children (29.2%) than among Russian children (72.2%, *p* = 0.002), as was their likelihood of achieving remission (Log-Rank test, *p* = 0.005), regardless of the JIA categories (*p* = 0.008). Sakha children had a 64.4% reduced chance of achieving remission on the first bDMARD, compared to Russian children (HR = 0.36, 95% CI: 0.18–0.71, *p* = 0.004). Conclusions: Distinct variations in the progression and treatment outcomes of JIA were observed between Sakha children and Caucasians. A tailored approach to the care of JIA patients is essential, considering their ethnic background.

## 1. Introduction

Juvenile idiopathic arthritis, or JIA, is the most prevalent chronic rheumatic condition affecting children [1]. JIA is a typical “umbrella diagnosis” that includes different categories of chronic arthritis with various kinds of articular and extra-articular features [2]. The current ILAR classification categorizes JIA; however, this remains a point of contention [3]. The geographic, ethnic, and genetic factors significantly influence the incidence, prevalence, and diversity of juvenile idiopathic arthritis (JIA) [4,5]. In Asian populations, there is a notably higher prevalence of certain rheumatic diseases, including spondyloarthritis, systemic lupus erythematosus, Takayasu’s arteritis, Behcet’s disease, and Kawasaki disease [6].

The indigenous peoples of the Arctic region, including the Sakha, as well as all the peoples of the Circumpolar world, possess a unique genetic code and a particular burden of hereditary diseases. They live in extreme climatic conditions and have restricted and monotonous diets [7,8,9,10].

Respiratory infections are among the most severe diseases affecting people living in circumpolar territories, primarily due to remoteness, isolation, cold weather, restricted dietary conditions, and specific immune system deficiencies [11,12,13]. Among adults in the Arctic, cerebrovascular disease is more prevalent, while the incidence of coronary heart disease is comparable to that of other regions [14]. Additionally, the prevalence of diabetes, obesity, arthritis, and suicide rates are similar to those observed in other populations [15,16,17,18,19].

The most common chronic conditions for Canadian Aboriginal populations were arthritis and high blood pressure [20]. The epidemiological studies in children of Sakha origin showed the following primary diseases in children: (1) Infections and respiratory system ailments; (2) gastrointestinal diseases (with caries predominance); (3) central nervous system disorders, followed by the musculoskeletal diseases [21,22].

Despite the high prevalence of musculoskeletal diseases in the indigenous population of the circumpolar region, detailed information, especially in children, is scarce. Our previous studies showed the higher (up to 7-fold time) prevalence of SLE in Sakha compared to the neighboring regions and the high prevalence of JIA in children [23,24]. A similar rate was shown in the Alaska native population [25].

### Objectives

We conducted the present study to describe the differences in JIA between the indigenous population and Caucasians living in the Arctic area, which may be important for the development of personalized approaches in diagnosis and treatment.

## 2. Materials and Methods

### 2.1. Design of the Study and Selection of Participants

This observational, retrospective cohort study analyzed the medical records of patients under eighteen years of age with JIA who were treated in the cardiorheumatology department of the Pediatric Center at Republican Hospital #1, M.E. Nikolaev Medical Center, between 2016 and 2023.

#### 2.1.1. Inclusion Criteria

(1)Diagnosis of JIA(2)Minors, or those under 18 years of age.

#### 2.1.2. Exclusion Criteria

Individuals may be diagnosed with systemic diseases other than juvenile idiopathic arthritis, which include systemic lupus erythematosus, vasculitis, connective tissue disorders, and various autoimmune inflammatory conditions. Every patient underwent a thorough assessment by a rheumatologist to either confirm or exclude the diagnosis of juvenile idiopathic arthritis (JIA) and identify the specific subtype of the condition. The diagnosis of JIA was established in accordance with the ILAR criteria [3]. Eye involvement was assessed through an ophthalmic examination using biomicroscopy.

### 2.2. Data Collection

The following information was revealed from the medical records:(a)Clinical and demographic information: Detailed patient information includes gender, date of birth, year of diagnosis, region of residence, ethnicity, family history, and the triggering factor of the illness. JIA subtype, age of JIA onset, number of active joints at onset, and presence of uveitis. The number of mixed families in the Sakha population is minimal, only 3%. According to the socio-cultural features, the question about ethnicities in mixed Sakha families is based on the mother’s opinion. The term “Russians” is more difficult, because it includes all white Caucasian peoples with self-identifications as Russian.(b)Laboratory features: baseline clinical blood count, erythrocyte sedimentation rate (ESR), C-reactive protein (CRP); presence of HLA-B27 antigen, antinuclear antibodies (ANA), rheumatoid factor (RF), antibodies against cyclic citrullinated peptides (anti-CCP), and levels of immunoglobulin classes A (IgA), M (Ig M), and G (Ig G).(c)Treatment options: We evaluated various antirheumatic treatments, including non-steroidal anti-inflammatory drugs (NSAIDs), systemic (oral and intravenous) and local (intra-articular injections) glucocorticosteroid therapy, non-biologic disease-modifying antirheumatic drugs (nbDMARDs) and biologic disease-modifying antirheumatic drugs (bDMARDs), as well as treatment duration.(d)Outcomes: attainment of juvenile idiopathic arthritis (JIA) remission according to C. Wallace criteria [26], the specific date of remission, any occurrences of JIA flare-ups, and the duration until the next flare-up.

### 2.3. Subgroup Analysis

We compared Sakha and Russian patients with JIA.

Statistics: The collected data underwent statistical analysis using R software, version 4.4.0, released on 24 April 2024, in conjunction with the RStudio environment, version 2024.04.2-764. Categorical data were presented using their absolute values. Independent categorical variables were assessed through 2 × 2 contingency tables, and Fisher’s exact tests were applied for statistical significance. Quantitative variables were tested for normality using the Shapiro-Wilk test and presented as the median (Me), with the interquartile range (IQR) shown as the first and third quartiles (25%, 75%). A paired t-test was applied to compare two independent quantitative variables if the data followed a normal distribution. Otherwise, the Mann-Whitney U-test was employed. Survival analysis with Kaplan-Meier curves and the Cox proportional hazards model was performed to estimate the probability of achieving the event (remission on methotrexate and initiation of bDMARDs). Likelihood ratio (LR), hazard ratio (HR), Wald, and Score tests were employed to assess the validity of the Cox proportional hazards model. The Schoenfeld residuals test was used to determine and verify the model’s assumption of proportional hazards. The missing data were excluded from the analysis. To prevent the issue of multiple comparisons, the Bonferroni correction was applied. All the statistical tests conducted were two-tailed, with a significance threshold set at *p* < 0.05.

## 3. Results

### 3.1. Differences in the Course of JIA Between Sakha and Caucasian Populations

The study included patients of Sakha (*n* = 168) and Russian (*n* = 48) nationality. The children of Sakha ethnicity had a later age of JIA onset (9.0 [IQR 6.0–12.0]) vs. 6.0 [4.0–11.0], *p* = 0.024) and a higher proportion of males (*n* = 85; 50.6% vs. *n* = 19; 38.8%, *p* = 0.198). Among the Sakha children, the enthesitis-related arthritis (ERA) category was predominant (*n* = 86; 51.2% vs. *n* = 9; 18.3%), whereas oligoarticular JIA prevailed among the Russian children (*n* = 26; 53.1% vs. *n* = 47; 27.9%, *p* = 0.0002). Sakha children exhibited higher rates of enthesitis (*n* = 32; 19% vs. *n* = 1; 2.0%, *p* = 0.003) and sacroiliitis (*n* = 40; 23.8% vs. *n* = 1; 2.0%, *p* = 0.0003) compared to Russian children. In the laboratory findings, only the HLA-B27 antigen exhibited a significantly higher prevalence among children from the Sakha population diagnosed with JIA (76/164, 46.3%, compared to 7/48, 14.6%, *p* = 0.00005). Other initial disease indicators, such as hemoglobin, leukocyte, and platelet counts, erythrocyte sedimentation rate (ESR), C-reactive protein (CRP), anti-cyclic citrullinated peptide (anti-CCP), rheumatoid factor (RF), and antinuclear antibody (ANA) levels, did not show significant differences between the ethnic groups. The data are summarized in Table 1.

### 3.2. Treatment of JIA in Studied Populations

There were no significant differences in the frequency of systemic or topical corticosteroid use, methotrexate, or biologic disease-modifying antirheumatic drug (bDMARD) administration between the groups. Methotrexate discontinuation was more common among the Sakha children (*n* = 24; 15.7%) compared to the Russian children (*n* = 4; 8.9%, *p* = 0.333) and was more frequently associated with poor tolerance and insufficient therapeutic response. The most commonly reported methotrexate-related adverse effects were nausea, vomiting, toxic hepatitis, and thrombocytopenia. The Sakha children were characterized by relatively high rates (*n* = 70; 41.7%) and delayed initiation of biological therapy. The Sakha children received adalimumab more frequently (*n* = 18; 25.7% vs. *n* = 1; 5.6%) compared to the Russians, while the Russian children were more commonly treated with tocilizumab (*n* = 3; 4.3% vs. *n* = 4; 22.2%, *p* = 0.027) compared to the Sakha children. The treatment was prescribed according to the national and international recommendations for JIA treatment in accordance with the drug labels. The time to bDMARD was twice as long in Sakha children, but the difference was insignificant (*p* = 0.158). Biologics were administered more frequently in the ERA JIA category in the Sakha children compared to the Russians (*n* = 46/86; 52.3% vs. *n* = 3/9; 33.3%; *p* = 0.0003). The remission rate was notably lower among the Sakha children (*n* = 19; 29.2%) compared to the Russian children (*n* = 13; 72.2%, *p* = 0.002). Similarly, the probability of achieving remission was significantly lower in the Sakha group (Log-Rank test, *p* = 0.005). Children from the Sakha ethnic group experienced remission infrequently, regardless of the categories of juvenile idiopathic arthritis (JIA) (*p* = 0.008). The Cox proportional hazards model revealed that children of Sakha origin had a 64.4% lower chance of remission with the first biologic DMARD compared to Russian children (HR = 0.36, 95% CI: 0.18–0.71, *p* = 0.004). The main changes in the biologic DMARD treatment were switches from etanercept to adalimumab due to primary or secondary inefficacy, and the development of a few cases of de novo uveitis. These findings are summarized in Table 2 and illustrated in Figure 1.

## 4. Discussion

The research is unique in that it focuses on juvenile arthritis in children residing in the Circumpolar region. Utilizing registry data, the study revealed variations in clinical and laboratory features, as well as treatment results, among JIA cases in Sakha children when compared to Russian children from the Republic of Sakha (Yakutia). While musculoskeletal disorders are prevalent among northern indigenous populations, affecting both youth and adults, research in this field remains scarce.

Research has shown that individuals belonging to northern indigenous First Nations are at an increased risk of developing rheumatoid arthritis [19] and ankylosing spondylitis [18].

Further research on the characteristics of juvenile arthritis among indigenous Arctic populations requires determining diagnostic, therapeutic, and preventive strategies for such conditions. Previous studies in adult populations have revealed a high prevalence of the HLA B27 antigen among healthy indigenous Arctic populations. This prevalence, commonly observed in Asian populations, suggests a shared genetic lineage and potential human migration from Asia to the Americas via the Beringia land bridge [27].

Juvenile idiopathic arthritis, specifically the enthesitis-related category, is the most common form observed in children from the Sakha region. This type of arthritis closely mirrors adult spondyloarthritis with respect to its clinical presentation and underlying pathology [24,28].

This type of arthritis differs in its etiological relationship to the preceding gastrointestinal infections. Our previous investigation demonstrated that gastrointestinal infections are the most prevalent triggers of reactive arthritis, resulting in enthesitis-related arthritis in the Sakha pediatric population, without any national differentiations [24]. In our cohort, the separate analysis has revealed that preceding gastrointestinal infections provoking JIA occurred in 11.3% of Sakha children and only 6.3% of Russians. It correlated with a lack of access to clean water and certain dietary habits, such as the consumption of fresh, frozen fish, meat, and animal internal organs (e.g., liver and intestine), which explains the relatively high frequency of preceding gastrointestinal infections in people living in rural circumpolar territories.

Infectious diseases constitute the predominant health issues affecting the population residing in the Circumpolar regions. This prevalence can be attributed to genetic predispositions, geographical isolation, and limited exposure to various pathogens. Exposure to a restricted diversity of pathogenic agents and human genetic diversity may increase susceptibility to immune-inflammatory disorders [29,30]. The ongoing climate change and subsequent warming in the Circumpolar region are precipitating the emergence of novel pathogens, which the immune systems of indigenous populations are not adapted to [31]. Microorganisms found in the Arctic and the Antarctic regions show a distinct antimicrobial resistance profile [32]. The influenza and COVID-19 pandemics have demonstrated a high mortality rate among isolated groups of the population. The limited intake of nutritional substances by indigenous people residing in the Arctic region leads to a dysbiotic gut microbiome, which significantly contributes to the development of immunoinflammatory diseases [33]. The indigenous people of the Arctic region face a higher prevalence of gastrointestinal diseases, frequent dental caries, and a significant rate of smoking. These factors contribute to an increased risk of immune-inflammatory disorders. Numerous medical studies have established a strong link between oral health issues, tobacco use, and the susceptibility to rheumatoid arthritis [34,35].

Among the environmental factors that significantly affect the pathogenesis of immuno-inflammatory diseases, it is essential to underline the contamination of soil, water, and food resources by heavy metals [36,37,38,39]. Rheumatoid arthritis among the indigenous population of the North has distinctive characteristics. Certain tribes demonstrate an increased prevalence of rheumatoid arthritis, with incidence rates reaching 2–7%, which is considered to be the highest worldwide. However, the data received from the other tribes do not confirm these findings [40,41,42,43]. Among the genetic risk factors associated with rheumatoid arthritis, the HLA-DRB1*1402 antigen is a distinctive allele that encodes shared epitopes and is found to be prevalent in indigenous native peoples in the US [44]. Environmental risk factors, such as tobacco smoking and periodontal diseases, prevalent among indigenous peoples of North America, are taken into account [34,35,45,46]. Early age of the first pregnancy and multiple pregnancies lead to a higher risk of rheumatoid arthritis in Indigenous North Americans (INA) [47]. An earlier onset characterizes rheumatoid arthritis in INA approximately 10 years younger than in non-Indigenous North Americans [40]. Another feature is that the indigenous population has limited access to comprehensive rheumatology care [48]. Early onset of arthritis and restricted access to medical care contribute to the progression of severe, end-stage joint pathology, resulting in surgical replacement of the affected joint [49], and premature mortality is high compared to non-INA, as is that of INAs with RA compared to non-INA with RA [50]. The incidence of active tuberculosis among individuals of Alaska Native and American Indian descent who have rheumatoid arthritis is 15.1%. This elevated prevalence significantly restricts the efficacy of contemporary antirheumatic treatments, including tumor necrosis factor-alpha (TNF-α) inhibitors [51]. Ankylosing spondylitis is also a frequent rheumatic disease in the INA. The incidence of ankylosing spondylitis among Alaskan Eskimos was 2.5% among adults over 20 years of age, with the same frequency among men and women [52]. It is significantly different from other countries and ethnic groups, where the incidence of ankylosing spondylitis ranges from 0.1% to 1.4%, and the ratio between men and women varies from 1.2 in Turkey to 7.0 in Italy [53]. Studies on JIA in children of the North and the Arctic Indigenous population are rare [24]. The research on epidemiological and clinical features of JIA in children of various ethnic groups is of particular interest. The current study revealed significant differences in clinical manifestations, laboratory parameters, and response to therapy among patients with JIA based on ethnicity. It is known that Asian populations, particularly the Sakha (~25%), are characterized by a high frequency of the HLA-B27 allele, which may explain the association between a larger number of cases of arthritis associated with enthesitis and sacroiliitis in the Sakha population, compared to Russians [9].

The lack of access to the healthcare system, lower parental education, and lower socioeconomic status of parents of indigenous patients with JIA, living in the rural Circumpolar territories, could be factors that disrupted the speed of high-quality medical service, and the possibility to target the window of opportunity in JIA treatment. The delayed diagnosis may be a significant factor that leads to an increased use of both non-biologic and biologic DMARDs. The prevalence of the HLA B27 antigen and enthesitis-associated arthritis is similar to that in many Asian populations. In several Asian countries, including India, Pakistan, and Singapore [54,55,56], enthesitis-related arthritis is the most prevalent subtype of juvenile idiopathic arthritis. This pattern is also observed in the Sakha population [57,58]. Children of Sakha have a high incidence of intestinal infections, which trigger arthritis. The urgent challenge of providing access to clean, uncontaminated water is a pressing concern for those residing in the Arctic region. A high rate of waterborne infections is correlated with a high incidence of post-enterocolitis cases of enthesitis-related arthritis associated with HLA-B27 antigens [59]. In Europe, oligoarthritis is responsible for up to 50% of all juvenile idiopathic arthritis (JIA) cases, while polyarthritis accounts for 20–30%. Systemic arthritis affects 10–15% of affected children, and psoriatic arthritis and enthesitis-related arthritis (ERA) each constitute 5–10% of the total JIA cases in this population [60].

A similar situation is described in Africa and the Middle East [61]. The highest frequency of ERA among Middle East countries is reported in Turkey, at 32.8–32.9% in the Western Anatolia region and Istanbul [62,63]. In contrast, other Turkish studies have found frequencies ranging from 10 to 21% [64,65,66]. Turkey also leads in the prevalence of JIA—64 per 100,000 children in the Middle East and Africa [67]. ERA is the most prevalent form of juvenile idiopathic arthritis (JIA) in several Asian countries, including India, Pakistan, and Singapore [54,55,56]. In Japan, systemic arthritis is the predominant variant [68]. A study conducted in Singapore found that ERA accounted for 32.8% of JIA cases, with a higher incidence among boys (60.6%). The HLA B27 antigen was frequently present in patients with ERA (79.8%), while uveitis was observed in only 2.8% of cases, aligning with previous findings [55]. In India, the HLA B27 antigen was most commonly associated with ERA (87%), and this association was linked to sacroiliitis. The antigen was also detected in children with oligoarthritis (10.3%), RF-positive polyarthritis (16%), RF-negative polyarthritis (26%), psoriatic arthritis (66%), and undifferentiated arthritis (40%). Notably, it was absent in patients with systemic JIA. Researchers highlighted the relatively low correlation between the HLA B27 antigen and typical ERA symptoms such as lower back pain, enthesitis, and sacroiliitis [69]. In the United Kingdom, recent studies have revealed ethnic disparities in JIA prevalence. The incidence among Asians and Africans was approximately 42.1 and 46.3 cases per 100,000 children, respectively, which was significantly lower than in Caucasians (71.1 per 100,000). The lowest prevalence was observed among children of mixed ethnicity (29.4 per 100,000) [70]. Interestingly, the prevalence of JIA in children of Sakha origin is higher than in Indian children from Alaska (79 per 100,000), as well as in many Asian countries, although it remains lower than in Germany, where it ranges from 133 to 168 per 100,000 [25,60].

Children of Sakha ethnicity experience a more severe clinical course of JIA. The HLA-B27 antigen is known to be linked to a more severe form of arthritis [71]. Children of the Sakha ethnicity experience a more aggressive and challenging-to-treat type of arthritis, even though the disease typically begins at a later age. Sakha children were less tolerant to methotrexate therapy, which might be associated with the studies of possible unique genetic variants of the cytochrome P450 genes in the Indigenous North American population, which led to the alteration of drug/xenobiotic detoxification [72]. Previous epidemiological studies have shown a higher frequency of polymorphic CYP2C19*2 and CYP2C19*3 alleles in the native Sakha people, similar to those in Asians (Southeast Asia and Oceania) [73]. The data from population-based pharmacogenomic analysis may be crucial for setting up a personalized treatment algorithm, realized through dose-correction strategies, closer side effect monitoring, or early switch to biologics in cases where methotrexate might be ineffective or intolerable.

Biologic therapy was less effective for this group of patients. Other studies have shown similar results; researchers from Taiwan [71] demonstrated that their cohort was also characterized by a persistent course of the disease, necessitating more aggressive treatment. Another study from Taiwan shows that HLA-B27-associated JIA subtypes are associated with more adverse outcomes [74]. These data may be used for future personalized treatment strategies, including the early administration of biological therapy in HLA-B27-positive patients with additional clinical, laboratory, or instrumental markers of poor outcomes. The Russian children in the Republic of Sakha predominantly suffered from oligoarthritis, which is characteristic of the Caucasian populations [60]. It suggests that genetic factors have a greater impact on the development of rheumatic diseases.

Sakha children received biological therapy later despite having a more severe course of arthritis compared to children living in other territories of the Russian Federation. The late administration of biological therapy is likely due to remote living and limited access to specialized medical care, which is often hindered by the lack of roads during the summer. Restricted access to healthcare for other indigenous peoples of the North has been discussed earlier [48]. The introduction of modern artificial intelligence-based systems for disease suspicion and web applications for remote disease monitoring may decrease the gap to specialized healthcare. Timely medical and treatment interventions could improve disease control, the safety of treatment, quality of life, and disease outcomes.

## 5. Study Limitations

The research’s limitations stem from its retrospective design. The study relies on existing documentation, which might contain incomplete or inaccurate details. The patients’ nationality was self-declared by their parents, potentially influencing the results, particularly in mixed families. The small sample size increases the risk of Type II errors, where actual effects might go undetected. The lack of standardized definitions for ethnicity identification could lead to data inconsistencies, affecting the study’s conclusions. Moreover, the remote location from specialized medical services, lower parental education, and socioeconomic challenges faced by indigenous patients with JIA in rural Circumpolar areas, characterized by harsh climates and logistical difficulties, could delay the initiation of specialized treatment and impact outcomes. This delay might contribute to the higher prevalence of non-biologic and biologic DMARDs, skewing the study’s findings. The researchers were unable to control the length of treatment or the specific medications given. The absence of a standardized diagnostic procedure and a unified treatment protocol may also have influenced the study’s outcomes. The study found correlations between ethnicity and JIA characteristics but could not establish causation. It did not take into account confounding factors that could affect results, such as the age at which symptoms first appeared, gender, pre-existing conditions, changes in treatment protocols over time, or variations in healthcare access and delays in treatment. These elements could distort the results and their interpretation. A more thorough analysis that accounts for these variables would provide more robust and trustworthy findings.

## 6. Conclusions

Distinct ethnic disparities in the manifestation of juvenile idiopathic arthritis (JIA) have been observed between Sakha children and the broader Russian pediatric population. Specifically, Sakha children exhibit a more severe and treatment-resistant form of the disease. This difference may be partly attributed to genetic and environmental influences that affect the development of the inflammatory process. Studying ethnic-specific features is of great importance, as it contributes to the development of prevention strategies, early diagnosis protocols, and access to specialized medical care for patients in remote areas of the Far North. A personalized approach to patient management, taking into account the patient’s ethnicity, is obligatory to optimize therapeutic outcomes.

The study is part of the scientific research work of the Yakut Science Center for Complex Medical Problems concerning “Fundamentals for development and health preservation in the pediatric population of the North” (state registration number 1022041300003-6), state assignment of the Ministry of Science and higher education of the Russian Federation (FSRG-2023-0003).

## Figures and Tables

**Figure 1 children-12-01525-f001:**
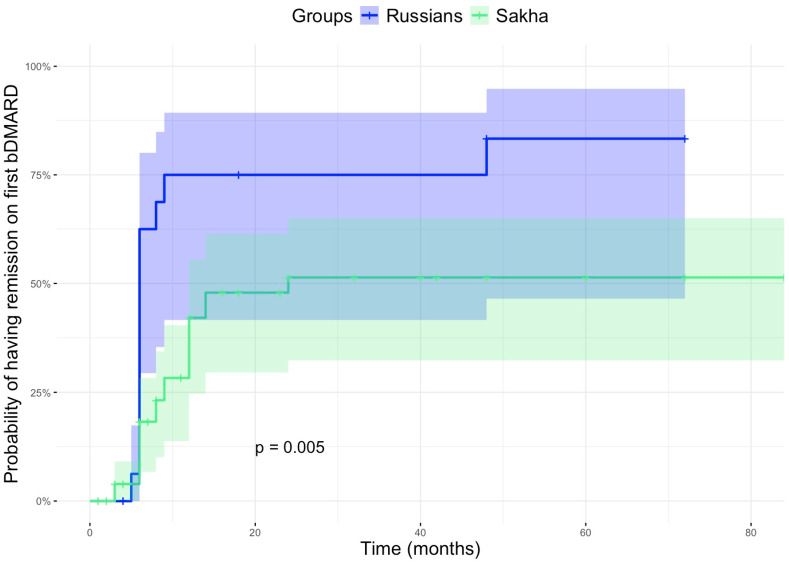
The probability of JIA remission in patients of different ethnicities treated with biologic disease-modifying antirheumatic drugs (bDMARDs).

**Table 1 children-12-01525-t001:** Demographic characteristics of patients with JIA of Sakha and Russian origins.

Parameter	Sakha Children, *n* = 168	Russian Children, *n* = 48	*p*-Value
Sex, male, *n* (%)	85 (50.6)	19 (38.8)	0.198
Age at onset *	9.0 [6.0–12.0]	6.0 [4.0–11.0]	0.024
Asian ethnicity, *n* (%)	168 (100.0)	0 (0.0)	
Place of residence, urban, *n* (%)	87 (51.8)	30 (61.2)	0.259
JIA category, *n* (%)			0.0002
-Oligoarthritis	47 (27.9)	26 (53.1)	
-RF-negative polyarthritis	25 (14.9)	8 (16.3)	
-RF-positive polyarthritis	2 (1.2)	0 (0.0)	
-Enthesitis-related arthritis	86 (51.2)	9 (18.3)	
-Systemic arthritis	3 (1.8)	4 (8.2)	
-Psoriatic arthritis	5 (3.0)	2 (4.1)	
Presence of enthesitis, *n* (%)	32 (19.0)	1 (2.0)	0.003
Presence of sacroiliitis, *n* (%)	40 (23.8)	1 (2.0)	0.0003
Presence of uveitis, *n* (%)	18 (10.7)	4 (8.2)	0.790
Presence of psoriasis, *n* (%)	5 (3.0)	2 (4.1)	0.658
Active joints *	4.0 [1.0–24.0]	3.0 [1.0–24.0]	0.328
RF positivity, *n* (%)	1 (0.6)	0 (0.0)	1.0
HLA-B27 positivity, *n* (%)	76/164 (46.3)	7/48 (14.6)	0.00005
ANA positivity, *n* (% among tested)	24/36 (66.7)	5/8 (62.5)	1.0
Hemoglobin, g/L *	119.0 [107.5–128.5]	122.0 [111.0–127.0]	0.322
Leukocytes, 10^9^/L *	7.8 [6.1–10.0]	7.6 [6.0–9.7]	0.744
Platelets, 10^9^/L *	379.0 [325.0–469.0]	350.0 [291.0–455.0]	0.081
ESR at onset, mm/h *	27.0 [1.0–117.0]	20.0 [0.0–65.0]	0.304
CRP at onset, mg/L *	4.4 [0.0–303.0]	3.0 [0.0–59.0]	0.645

Note: * (Me, [IQR]) Abbreviations: ANA—antinuclear antibodies, CRP—C-reactive protein, ESR—erythrocyte sedimentation rate, RF—rheumatoid factor.

**Table 2 children-12-01525-t002:** Treatment of JIA and the outcomes in the patients of Sakha and Russian origins.

Parameter	Sakha Children, *n* = 168	Russian Children, *n* = 48	*p*-Value
**Therapy**			
-No corticosteroids	132 (78.6)	41 (83.7)	0.886
-Intravenous corticosteroids	9 (5.4)	2 (4.1)	
-Oral corticosteroids	3 (1.8)	1 (2.0)	
-Intra-articular corticosteroids	18 (10.7)	3 (6.1)	
-Intravitreal corticosteroids	6 (3.6)	2 (4.1)	
Methotrexate	153 (91.6)	45 (91.8)	1.0
Methotrexate discontinuation	24 (15.7)	4 (8.9)	0.333
Patients treated with bDMARD	70 (41.7)	18 (36.7)	0.621
First bDMARD, *n* (%)			0.027
-Adalimumab	18 (25.7)	1 (5.6)	
-Abatacept	1 (1.4)	0 (0.0)	
-Secukinumab	1 (1.4)	1 (5.6)	
-Tocilizumab	3 (4.3)	4 (22.2)	
-Etanercept	47 (67.2)	12 (66.6)	
Time to bDMARD initiation, months *	8.0 [0–60.0]	4.0 [0–60.0]	0.153
The frequency of bDMARD administration, according to JIA categories			0.0003
-Oligoarthritis	8/47 (17.0)	7/24 (29.2)	
-RF-negative polyarthritis	9/25 (36.0)	4/8 (50.0)	
-RF-positive polyarthritis	0/2 (0.0)	0/0 (0.0)	
-Enthesitis-related arthritis	45/86 (52.3)	3/9 (33.3)	
-Systemic arthritis	3/3 (100.0)	4/4 (100.0)	
-Psoriatic arthritis	4/5 (80.0)	0/2 (0.0)	
**JIA outcomes**			
Remission on first bDMARD, *n* (%)	19 (29.2)	13 (72.2)	0.002
The achievement of remission on first bDMARD, according to JIA categories			0.008
-Oligoarthritis	1/7 (14.3)	4/7 (57.1)	
-RF-negative polyarthritis	3/8 (37.5)	3/4 (75)	
-RF-positive polyarthritis	0/0 (0.0)	0/0 (0.0)	
-Enthesitis-related arthritis	13/43 (30.2)	2/3 (66.6)	
-Systemic arthritis	1/3 (33.3)	4/4 (100)	
-Psoriatic arthritis	1/3 (33.3)	0/0 (0.0)	
Time to remission on the first bDMARD *	11.0 [1.0–84.0]	6.0 [4.0–72.0]	0.171

Note: * (Me, [IQR]) Abbreviations: bDMARD—biological disease-modifying antirheumatic drug, JIA—juvenile idiopathic arthritis, RF—rheumatoid factor.

## Data Availability

The datasets generated during and/or analyzed during the current study are available from the corresponding author upon reasonable request.

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
