# Peer review of "Ethnic Differences in Juvenile Idiopathic Arthritis in the Circumpolar Region"

_children, 2025, doi:10.3390/children12111525_

Round 1
Reviewer 1 Report
Comments and Suggestions for Authors
This study examined ethnic differences in juvenile idiopathic arthritis (JIA) among indigenous Sakha and Caucasian (Russian) children living in Yakutia, an Arctic region with unique environmental conditions. Using a retrospective cohort design, data from 216 patients (168 Sakha and 48 Russians) treated at a pediatric rheumatology center between 2016 and 2023 were analyzed. Results showed that Sakha children experienced later disease onset and were more frequently affected by the enthesitis-related arthritis (ERA) category, enthesitis, sacroiliitis, and HLA-B27 positivity compared to Russian peers. Biologic therapy initiation was delayed in Sakha patients, contributing to lower remission rates (29.2% vs. 72.2%) and reduced likelihood of remission on first-line biologics. These findings indicate substantial ethnic variation in JIA phenotype, disease progression, and treatment outcomes. The study highlights the importance of considering ethnicity when diagnosing and managing JIA and underscores the need for personalized therapeutic strategies tailored to specific populations.
Altogether this is an important and timely article, this reviewer has certain suggestions that would help produce a more comprehensive overview of the topic:
Comments:
1, The introduction highlights the high prevalence of rheumatic diseases in Asian populations but could benefit from a more detailed discussion of why ethnic and geographic differences in JIA are clinically important.
2, Please expand on the global literature about ethnic variations in JIA (e.g., data from East Asian, Native American, or other Arctic populations) to situate your study within broader research.
3, The remission analysis (Log-Rank test, HR, CI) is appropriate but would benefit from Kaplan-Meier curves in the main text or supplementary material.
4, Please clarify whether adjustments were made for potential confounders (e.g., age at onset, sex, treatment delay) in the Cox regression analysis.
5, Future directions should consider the role of pharmacogenomics and healthcare equity in personalized treatment strategies.
6, A brief description of the healthcare context in Yakutia would help international readers understand systemic factors influencing treatment initiation.
Author Response
Reviewer 1
This study examined ethnic differences in juvenile idiopathic arthritis (JIA) among indigenous Sakha and Caucasian (Russian) children living in Yakutia, an Arctic region with unique environmental conditions. Using a retrospective cohort design, data from 216 patients (168 Sakha and 48 Russians) treated at a pediatric rheumatology center between 2016 and 2023 were analyzed. Results showed that Sakha children experienced later disease onset and were more frequently affected by the enthesitis-related arthritis (ERA) category, enthesitis, sacroiliitis, and HLA-B27 positivity compared to Russian peers. Biologic therapy initiation was delayed in Sakha patients, contributing to lower remission rates (29.2% vs. 72.2%) and reduced likelihood of remission on first-line biologics. These findings indicate substantial ethnic variation in JIA phenotype, disease progression, and treatment outcomes. The study highlights the importance of considering ethnicity when diagnosing and managing JIA and underscores the need for personalized therapeutic strategies tailored to specific populations.
Reply:
Dear Reviewer! We sincerely appreciate your positive feedback and thoughtful review. Our answers (A) to your queries (Q) are below and highlighted by color in the manuscript.
Altogether this is an important and timely article, this reviewer has certain suggestions that would help produce a more comprehensive overview of the topic:
Comments:
Q1. The introduction highlights the high prevalence of rheumatic diseases in Asian populations but could benefit from a more detailed discussion of why ethnic and geographic differences in JIA are clinically important.
A1. Dear Reviewer! This information has been added to the discussion section.
Q2. Please expand on the global literature about ethnic variations in JIA (e.g., data from East Asian, Native American, or other Arctic populations) to situate your study within broader research.
A2. Dear Reviewer! The information about the distribution and ethnic variations in JIA has been added now.
Q3. The remission analysis (Log-Rank test, HR, CI) is appropriate, but would benefit from Kaplan-Meier curves in the main text or supplementary material.
A3. Dear Reviewer! In the manuscript, a Kaplan-Meier curve is presented, using remission analysis as the event of interest. Our analysis includes only the remission on biologics. We do not have data on overall remission rates, particularly for non-compliance with medication.
Q4. Please clarify whether adjustments were made for potential confounders (e.g., age at onset, sex, treatment delay) in the Cox regression analysis.
A4. Dear Reviewer! We did not include the confounders in the Cox regression, which you mentioned above. The statement about it has been included now in the Limitations section.
Q5. Future directions should consider the role of pharmacogenomics and healthcare equity in personalized treatment strategies.
A5. Dear Reviewer! We absolutely agree with you! Population-based pharmacogenomic studies are needed, for example, to assess genes associated with xenobiotic detoxification, which may change the treatment protocol by using low doses of methotrexate or early switching to biologics. Even assessment of HLA B27 with high lab inflammation could be an indication for early biologic administration. The following studies are needed. These statements have been added to the discussion now.
Q6. A brief description of the healthcare context in Yakutia would help international readers understand systemic factors influencing treatment initiation.
A6. Dear Reviewer! The main obstacles to early treatment are associated with living in a rural territory with low access to healthcare providers, especially for specialized medical care, such as rheumatology. Vast territory, the majority of arctic settlements are far away from medical centers (thousands of kilometers). During the warm time of the year, there are no car roads (only winter roads in snow), so the plane or helicopter is the universal means of communication. However, both are weather-dependent and expensive, despite government donations. It is easy to cross the rivers only in the winter on ice, and very difficult during the ice-breaking period and in the warmer part of the year. Telehealth systems and medical teams that come to far rural settlements have improved the situation, but new contemporary tools, as AI in medicine, might help in faster decision-making and faster transferring of the patient to specialized healthcare. Additionally, a more functional patient-monitoring system, combined with a disease-specific web application, could shorten the time to new treatment if the initial treatment fails. A summary of the items mentioned above has been added to the Discussion section.
Dear Reviewer!
Thank you so much for your help and efforts.
I hope the manuscript has improved after your suggestions.
On behalf of the Authors
Mikhail Kostik, MD, PhD, Professor
Reviewer 2 Report
Comments and Suggestions for Authors
This manuscript presents an important and timely study on ethnic differences in juvenile idiopathic arthritis (JIA) among children in the Republic of Sakha (Yakutia). The research contributes valuable insights into the underexplored area of rheumatic diseases in Arctic indigenous populations. However, several areas require clarification, strengthening of the discussion, and improvements in presentation before publication.
- The study provides meaningful data from an underrepresented population and highlights ethnic disparities in JIA presentation and outcomes. However, the manuscript should better emphasize what makes these findings novel compared to prior work by the same group (e.g., reference 24, World J Clin Pediatr 2025).
- Consider expanding on how missing data were handled, particularly for laboratory and clinical variables.
-
The analysis is generally sound and clearly presented. However, the authors should specify whether multiple comparison corrections were applied given the large number of p-values reported.
-
The discussion successfully integrates genetic and environmental hypotheses. Still, causal language (e.g., “the HLA-B27 antigen explains the large number of ERA cases”) should be tempered—these are associations, not mechanistic proofs.
-
The claim that methotrexate intolerance among Sakha children may be genetically mediated (via CYP450 differences) is interesting but speculative. The authors should either provide supporting data or frame it more cautiously.
-
The discussion would benefit from comparing these results with non-Arctic Asian and European JIA cohorts, to position the findings in a broader epidemiological context.
Author Response
Reviewer 2
This manuscript presents an important and timely study on ethnic differences in juvenile idiopathic arthritis (JIA) among children in the Republic of Sakha (Yakutia). The research contributes valuable insights into the underexplored area of rheumatic diseases in Arctic indigenous populations. However, several areas require clarification, strengthening of the discussion, and improvements in presentation before publication.
Reply:
Dear Reviewer! We sincerely appreciate your positive feedback and thoughtful review. Our answers (A) to your queries (Q) are below and highlighted by color in the manuscript.
Q1. The study provides meaningful data from an underrepresented population and highlights ethnic disparities in JIA presentation and outcomes. However, the manuscript should better emphasize what makes these findings novel compared to prior work by the same group (e.g., reference 24, World J Clin Pediatr 2025).
A1. Dear Reviewer!
In the previous manuscript, we emphasized the prevalence of JIA in the Republic of Sakha without clinical differences between the two main populations, living in Sakha (Sakha and Russians). We excluded from the analysis other minor nationalities residing in the Sakha Republic. In this manuscript, the primary objective was to compare the clinical phenotypes of JIA patients from two nationalities. The research on epidemiological and clinical features of JIA in children of various ethnic groups is of particular interest. The current study revealed significant differences in clinical manifestations, laboratory parameters, and response to therapy among patients with JIA based on ethnicity. This clarification has been added to the discussion section.
Q2. Consider expanding on how missing data were handled, particularly for laboratory and clinical variables.
A2. Dear Reviewer! The missing data were not included in the analysis. This statement was added in the statistics section.
Q3. The analysis is generally sound and clearly presented. However, the authors should specify whether multiple comparison corrections were applied given the large number of p-values reported.
A3. Dear Reviewer! The multiple comparison was applied only for the JIA distribution. The p-value is very, very low, so the multiple comparison correction test (Bonferroni) remains statistically significant—the statement about multiple comparison added in the Statistical section.
Q4. The discussion successfully integrates genetic and environmental hypotheses. Still, causal language (e.g., “the HLA-B27 antigen explains the large number of ERA cases”) should be tempered—these are associations, not mechanistic proofs.
A4. Dear Reviewer! Thank you so much for this comment—the modification to the sentence has been made. The new version is “It is known that Asian populations, especially the SIt is known that Asian populations, especially Sakha (~25%), are characterized by a high frequency of the HLA-B27 allele, which may explain the association with a larger number of cases of arthritis and sacroiliitis associated with enthesitis in the Sakha population, compared to Russians”.
Q5. The claim that methotrexate intolerance among Sakha children may be genetically mediated (via CYP450 differences) is interesting but speculative. The authors should either provide supporting data or frame it more cautiously.
A5. Dear Reviewer! There are no direct studies about the molecular mechanisms of methotrexate intolerance in Sakha children. Previous epidemiological studies have shown a higher frequency of polymorphic CYP2C19*2 and CYP2C19*3 alleles, similar to those found in Asians (South-East Asia and Oceania), associated with poor drug metabolism. The reference was also added. The data has a speculative nature.
Q6. The discussion would benefit from comparing these results with non-Arctic Asian and European JIA cohorts to position the findings in a broader epidemiological context.
A6. Dear Reviewer! The information has been added to the discussion section.
Dear Reviewer!
Thank you so much for your help and efforts.
I hope the manuscript has improved after your suggestions.
On behalf of the Authors
Mikhail Kostik, MD, PhD, Professor
Round 2
Reviewer 2 Report
Comments and Suggestions for Authors
The authors revised the manuscript accordingly.